# Encrypted Network Traffic Analysis and Classification Utilizing Machine Learning

**DOI:** 10.3390/s24113509

**Published:** 2024-05-29

**Authors:** Ibrahim A. Alwhbi, Cliff C. Zou, Reem N. Alharbi

**Affiliations:** Department of Computer Science, University of Central Florida, Orlando, FL 32816, USA; ibrahim.alharbi@ucf.edu (I.A.A.); re065413@ucf.edu (R.N.A.)

**Keywords:** encrypted network traffic, machine learning, traffic classification, device fingerprinting

## Abstract

Encryption is a fundamental security measure to safeguard data during transmission to ensure confidentiality while at the same time posing a great challenge for traditional packet and traffic inspection. In response to the proliferation of diverse network traffic patterns from Internet-of-Things devices, websites, and mobile applications, understanding and classifying encrypted traffic are crucial for network administrators, cybersecurity professionals, and policy enforcement entities. This paper presents a comprehensive survey of recent advancements in machine-learning-driven encrypted traffic analysis and classification. The primary goals of our survey are two-fold: First, we present the overall procedure and provide a detailed explanation of utilizing machine learning in analyzing and classifying encrypted network traffic. Second, we review state-of-the-art techniques and methodologies in traffic analysis. Our aim is to provide insights into current practices and future directions in encrypted traffic analysis and classification, especially machine-learning-based analysis.

## 1. Introduction

Traffic analysis, a methodical examination of network activity, plays a vital role in various domains, offering insights into patterns and anomalies within network traffic [1]. However, the advent of network traffic encryption has posed formidable challenges to traditional analysis methods, rendering plaintext payload extraction less effective [2,3]. Machine learning has emerged as a potent solution, capable of extracting valuable insights from encrypted traffic analysis without accessing content [4]. This paradigm shift allows for the development of sophisticated algorithms that discern patterns and classify encrypted traffic accurately, overcoming the limitations posed by encryption protocols [5].

### 1.1. Utilize Machine Learning in Cybersecurity

Machine learning (ML) is a branch of artificial intelligence (AI) that empowers computers to learn and improve from experience without being explicitly programmed. It involves the development of algorithms that enable systems to automatically learn and make predictions or decisions based on data. At its core, ML algorithms leverage statistical techniques to identify patterns and relationships within datasets, allowing them to generalize from past examples and make accurate predictions on unseen data.

In the context of cybersecurity, machine learning plays a crucial role in enhancing threat detection, risk assessment, and anomaly detection. By analyzing vast amounts of data, ML algorithms can identify patterns indicative of malicious activity, enabling the early detection and mitigation of security threats. ML techniques are particularly effective in cybersecurity due to their ability to process large-scale data in real time and adapt to evolving threats.

We summarize the main cybersecurity application areas that people can utilize ML to enhance cybersecurity defense and effectiveness:Threat Detection: ML algorithms can analyze network traffic, system logs, and user behavior to identify abnormal patterns indicative of security threats such as malware infections, intrusion attempts, and unauthorized access.Anomaly Detection: ML models can learn the normal behavior of systems and users and detect deviations from this baseline, signaling potential security breaches or anomalies. This approach is particularly useful for detecting zero-day attacks and insider threats.Predictive Analysis: ML algorithms can predict potential security incidents based on historical data and ongoing trends, enabling proactive risk management and mitigation strategies.Vulnerability Management: ML techniques can be used to identify vulnerabilities in software and systems by analyzing code, network configurations, and historical data, helping organizations prioritize and remediate security weaknesses.Behavioral Analysis: ML algorithms can analyze user behavior, application usage patterns, and system interactions to identify suspicious activities and detect advanced persistent threats (APTs) that evade traditional security measures

### 1.2. Encrypted Network Traffic Analysis and Classification

Statistical techniques for detecting cyberattacks on computer networks based on an analysis of abnormal traffic behavior” were published in the International Journal of Computer Network and Information Security in 2020. This paper likely discusses various statistical methods used to detect cyberattacks by analyzing abnormal traffic behavior on computer networks.

In the context of “Encrypted Network Traffic Analysis and Classification”, this reference could be relevant in several ways. First, it may discuss statistical techniques used in the analysis of network traffic, including encrypted traffic. These techniques could involve identifying patterns or anomalies in encrypted network traffic to detect potential cyberattacks or security threats.

Furthermore, the paper might delve into how the encryption of network traffic poses challenges for traditional methods of network traffic analysis and classification. Encrypted traffic can obscure important information, making it difficult to detect malicious activities or classify network traffic accurately. Therefore, understanding statistical techniques for analyzing encrypted traffic and detecting abnormalities becomes crucial in ensuring network security.

Overall, this reference could provide insights into statistical methods used for analyzing encrypted network traffic, detecting cyberattacks, and how the insecurity introduced by encryption affects the analysis and classification of network traffic.

### 1.3. Organization of This Paper

The rest of this paper is organized as follows. In Section 2, we give a comprehensive introduction to prior survey and research work on network traffic classification. In Section 3, we introduce the three major applications that this paper focuses on for encrypted traffic analysis. In Section 4, we introduce the detailed procedure for utilizing machine-learning methodology in classifying network traffic. In Section 5, we summarize and explain the general network traffic analysis and inspection. We introduce the main strategies for analyzing encrypted traffic in Section 6. The unresolved challenges and limitations of current research in classifying encrypted network traffic are described in Section 7 and Section 8. We provide our recommendations and possible future perspectives in this research direction. Section 9 concludes this paper.

## 2. Related Work

This section serves to introduce relevant surveys and some important research studies on network traffic analysis and classification. Velan et al. [6] presented comprehensive definitions and information regarding commonly employed traffic encryption techniques. Prior to 2014, it also conducted research on the categorization of encrypted data. On the other hand, our objective is to thoroughly examine previous studies to assess the many techniques for detecting that have arisen in the past few years.

Conti et al. [7] conducted a comprehensive survey on the current state of network traffic analysis produced by mobile devices. They propose three criteria for a systematic classification of existing works: the goal of traffic analysis, the location of network traffic capture, and the specific mobile platforms being targeted. Zhang et al. [8] conducted a thorough analysis of the intersections between deep learning and mobile and wireless networks, effectively connecting these two domains. The authors conducted a comprehensive assessment of nearly 600 research publications and examined methods and platforms that improve the implementation of deep learning in mobile environments.

Wang et al. [9] conducted research on the utilization of deep-learning techniques for classifying encrypted traffic in mobile services. Their study specifically examined dataset selection, model input design, and model architecture. The authors further introduced a comprehensive system for classifying mobile encrypted traffic using deep-learning techniques. In addition, they identified some significant issues and obstacles with the application of deep learning in encrypted traffic classification. While these studies cover both encrypted and unencrypted traffic, they lack emphasis on detecting malicious traffic and mostly concentrate on mobile devices. Our review paper encompasses studies pertaining to the detection of encrypted harmful traffic while also emphasizing platforms beyond mobile devices.

Berman et al. [10] conducted an extensive analysis of deep-learning applications in the field of cyber security, encompassing several areas such as network traffic identification, network intrusion detection, malware classification, and more. In addition, the authors included concise explanations of various deep-learning algorithms, including restricted Boltzmann machines, recurrent neural networks (RNN), and generative adversarial networks (GAN). They also discussed a diverse array of attack categories, including malware, botnets, and spam.

Abbasi et al. [11] conducted a comprehensive examination of deep-learning techniques used in the field of network traffic monitoring and analysis (NTMA). Identical to the study conducted by [10], the authors furnished us with comprehensive explanations and essential context regarding deep-learning methods, including Multi-layer perceptron (MLP), Convolutional neural networks (CNN), Long short-term memory (LSTM), Auto-encoder (AE), and GAN models.

Aceto and colleagues [12] provided a comprehensive summary of the main topics in traffic analysis that are anticipated to be appealing for deep learning. The authors additionally presented a methodical taxonomy and classified the current traffic classifications based on deep learning. They presented a comprehensive deep-learning framework for classifying encrypted and mobile traffic. The framework includes a precise definition of the key components, such as the traffic object, types of input data, classification tasks, and deep-learning architecture.

Zhai et al. [13] introduced a system called the ’six-step technique’ for detecting encrypted malicious traffic. The authors also examined the current deep-learning-based methods for detecting encrypted malicious communications, using the suggested framework as a basis. Additionally, in [14], 20 preexisting public datasets are categorized and analyzed according to their relevant use cases, highlighting their advantages and disadvantages. Our study offers a thorough examination of both deep-learning and classical machine-learning methodologies, as well as a full analysis of features.

Tahaei et al. [15] conducted a survey in 2020 on the topic of traffic classification in Internet-of-things networks. In 2020, Salman et al. [16] conducted a comprehensive analysis of machine-learning methods used for Internet traffic classification. The study focused on various goals of traffic classification, including protocol classification, application classification, action classification, category classification, and device classification. AlDaajeh [17] conducted a comprehensive survey on the application of deep learning in wireless and mobile networking.

Several well-established technologies and techniques for identifying and categorizing unencrypted data, such as deep packet inspection (DPI) methods based on payload analysis and identification methods based on ports, already exist [18]. Nevertheless, due to the implementation of traffic encryption techniques, these conventional approaches are no longer viable. Furthermore, DPI techniques have raised issues regarding the privacy of user data.

Conversely, port-based traffic identification systems operate under the assumption that applications utilize established Transmission Control Protocol/User Datagram Protocol (TCP/UDP) port numbers designated by the Internet Assigned Numbers Authority (IANA). Consequently, when apps deviate from IANA rules by utilizing dynamic ports or encryption protocols like Peer to Peer (P2P) protocol, the methods used to identify traffic and applications based on ports become ineffective [19]. Various technologies are available for implementing traffic encryption, including SSL, Transport Layer Security (TLS), Virtual Private Network (VPN), Secure Shell Protocol (SSH), and P2P. Encryption algorithms vary in their operation, with some functioning at the transport layer and others at the application layer.

This distinction poses a significant challenge when it comes to classifying encrypted communications. The encrypted traffic exhibits diverse data distribution characteristics, even when employing the same encryption algorithm, owing to variations in the distribution and utilization of the original traffic. Therefore, the majority of research is centered on the binary categorization of encrypted traffic, namely the task of distinguishing harmful traffic from valid traffic [20,21].

Flow-based machine-learning techniques have emerged as the predominant way of classifying encrypted information. Nevertheless, it is important to highlight that the gathering of training datasets and the use of features for detecting encrypted communication are still subjects of intense investigation [22,23,24,25,26].

Researchers have recently suggested numerous detection techniques utilizing machine learning, which may be broadly categorized as traditional machine learning and deep learning. Traditional machine learning can be categorized into two distinct subcategories: supervised learning and unsupervised learning. In one study [27], Adnan et al. utilized three supervised learning algorithms, namely Random Forest (RF), Support Vector Machine (SVM), and XGBoost, to differentiate between fraudulent and legal HTTPS traffic. In another study [28], Agrawal et al. performed comparison experiments utilizing seven distinct supervised learning algorithms, including k-nearest neighbors (KNN), Classification And Regression Trees (CART), and Naïve Bayes.

The objective was to identify fraudulent traffic from a dataset containing multiple encryption protocols. In their 2021 study, Ma et al. [29] introduced an improved KNN algorithm for training a model that detects encrypted traffic. This technique enhances the estimation of KNN distances [30,31,32]. In the context of unsupervised learning, ref. [33] introduced an enhanced density peaks clustering approach to improve the precision and effectiveness of detecting encrypted malicious communications.

Celik et al. [34] conducted a performance comparison of the K-means, one-class support vector machine (OCSVM), least squares anomaly detection (LSAD), and KNN algorithms. They evaluated the algorithms using tamper-resistant features, including Goodput and the ratio between the maximum packet and minimum packet. Zhang et al. [35] introduced a new clustering approach that utilizes the harmonic mean to measure the distance between clusters, with the aim of identifying encrypted data.

Regarding the investigation of deep learning in detecting encrypted traffic, the primary focus of researchers lies in examining the efficacy of Convolutional Neural Networks (CNN), Recurrent Neural Networks (RNN), and Autoencoders (AE). Khalife et al. [36] presented a thorough framework for network traffic classification. The process of traffic detection and classification was divided into three distinct stages: data input, selection of classification approach, and output of final results. They provided a general overview of the existing technology and procedures for each level. During the step of selecting a classification approach, there are various ways available, including machine-learning technologies, statistical methods (such as heuristics), and graphical technique methods (such as motifs and graphlets) [37].

As shown in Table 1, our survey provides a comprehensive overview of machine-learning-based methods for analyzing and classifying encrypted network traffic.

## 3. Encrypted Traffic Analysis for Three Major Applications

In the current cyber world, there are many applications involving encrypted network traffic that need to be analyzed and classified by their stakeholders. However, from the perspective of technological merit and broader impact, in this paper, we focus on providing analysis and survey of three major applications: the Internet of Things, mobile devices and applications, and web applications.

### 3.1. The Internet-of-Things

The Internet-of-Things (IoT) encompasses a diverse range of interconnected objects, from industrial systems to household devices, interconnected through embedded electronics and software, facilitating data exchange and analysis among these entities [44]. Emerging IoT applications span various sectors, including healthcare, agriculture, and military, presenting new challenges in device management, data handling, and security [45,46]. Extensive research addresses architecture, communication protocols, and security mechanisms to ensure the successful commercialization of IoT technologies [47], with the utilization of empowering technologies like cloud computing and edge computing amplifying both opportunities and risks [48,49], necessitating robust security measures to safeguard against potential threats. Figure 1 illustrates the high-level cryptographic implementation in IoT system to ensure data integrity and authenticity.

The advent of encrypted network traffic poses significant challenges for analysis and classification, but leveraging machine-learning techniques offers promising solutions. The IoT plays a pivotal role in this landscape, facilitating vast data generation from interconnected devices and systems. Machine learning, particularly deep-learning models, proves invaluable in processing and extracting insights from the sensor data produced by IoT devices, aiding in anomaly detection and predictive maintenance [50]. Furthermore, machine-learning algorithms can enhance user experience on IoT-enabled websites by personalizing content based on user behavior and preferences, while natural language processing models improve user interaction [51,52]. In the realm of image and video processing, deep learning excels in tasks like facial recognition and object detection, enhancing security and user engagement [53]. Additionally, machine-learning techniques such as anonymization and encryption play crucial roles in ensuring data security, particularly in the exchange of sensitive information between IoT devices, websites, and applications [29,54,55]. As the volume of data continues to grow, scalability becomes imperative for deep learning and machine-learning architectures to effectively analyze encrypted network traffic and classify patterns, ensuring robust security measures in an increasingly encrypted landscape.

### 3.2. Mobile Devices and Applications

Since the invention of the iPhone, mobile devices and mobile applications have quickly grasped both our daily lives and the world of business. Mobile devices remove the physical barriers between people, connecting anyone instantly through calls, texts, video chats, social media apps, etc. They enable Internet access and information gathering at any place or time, significantly enhancing people’s daily lives and business transactions. Because of the importance of mobile devices and applications, attackers have developed many ways to extract information and user privacy via captured mobile devices’ encrypted traffic. Such attacks become more challenging since attackers can easily eavesdrop on the network traffic between mobile devices and the Internet via WiFi or cellular wireless channels.

Ahlgren et al. [56] showcased how encrypted traffic patterns analysis can discreetly deduce sensitive personal information, even over encrypted WiFi networks, without requiring network access credentials. While achieving a commendable recall rate of 86%, indicating potential inadvertent disclosure of personal information by applications, the experiment also reveals room for improvement, with a significant false positive rate. Furthermore, the techniques employed for app identification extend to fingerprinting other encrypted communications like VoIP and website visits, making mobile apps particularly susceptible targets due to their transparent ranking system and simplicity of data collection [57]. Notably, these techniques, demonstrated using standard 802.11 g WiFi, are anticipated to be applicable across various wireless communication protocols, including long-distance protocols like 4G LTE used in cellular networks [58]. The scenario examined involves a passive observer seeking to deduce information about users connected to a WiFi Access Point (AP), highlighting the potential implications of encrypted network traffic analysis utilizing machine learning to uncover sensitive information without compromising network security.

### 3.3. Website Fingerprinting

López et al. [59] demonstrated that an assailant can discern the specific webpage a client is visiting using a classification algorithm, which processes observed packet sequences. The crux of the classification lies in the concept of distance between packet sequences, where a greater distance suggests that the sequences are less likely to originate from the same webpage. Since then, various metrics for calculating distance have been employed by different researchers, including comparing the frequency of unique packet lengths or adaptations of the Levenshtein distance. The chosen distance metric reflects the approach to utilizing features to differentiate between web pages. These features are derived, whether directly or indirectly, from the packet sequences to enable comparative analysis.

The foundational insight presented by [60] showed that the webpage category is inherently multi-modal. Numerous variables can induce variations in a webpage: the network state, random changes in advertisements and content, updates over time, and the unpredictable sequence of resource loads. Even the client setup might influence how the page is rendered. To manage such multi-modal datasets, an attacker should amass sufficient data to cover representative samples from each variation. For instance, an attacker might collect data reflecting two different loading scenarios of a webpage: one under low-bandwidth and another under high-bandwidth conditions. The classifier we employ is tailored to handle multi-modal classes, allowing for the different modes within a category to be unrelated to each other.

Another category of website fingerprinting attack is based on resource size. The HTTP 1.0 specification mandates that individual resources of a web page, such as images, scripts, etc., be fetched over distinct TCP connections. This allows an attacker to distinguish between these various connections and deduce the total size of each resource. The initial research into this type of vulnerability was conducted in the era of HTTP 1.0, with Cheng et al., in 1998 [61], Sun et al., in 2002 [62], and Hintz in 2003 [63] demonstrating that knowledge of resource sizes could facilitate webpage identification. However, the subsequent HTTP 1.1 standard, which employs persistent connections, alongside advancements in browsers and privacy-protection technologies, has rendered such resource size-based attacks less effective against more recent systems.

## 4. Procedure of Machine-Learning-Based Classification of Network Traffic

In this section, we systematically introduce the major steps and the procedure in network traffic analysis and classification by utilizing the machine-learning method. The high-level architecture is shown in Figure 2 [14,64,65,66,67,68]. In the following, we explain each block of the procedure in detail.

*Data collection*: The landscape of Internet traffic is constantly changing due to the advent of novel forms of traffic, devices, and applications. Hence, it is necessary to gather well-recognized Internet traffic while taking into account emerging trends in dealing with unknown traffic. Much earlier work on traffic classification relies on capturing non-encrypted network traffic, which is not suitable for analyzing encrypted traffic for most Internet applications presently.

Lalmuanawma et al. [69] showed that most of the accessible public datasets lack labels or were labeled using unreliable techniques, such as deep packet inspection, which is ineffective with encrypted traffic, or port-based labeling, which is ineffective with dynamically allocated ports.

Data collection could have very different formats depending on where it is collected, such as the data-link frames captured on an encrypted WiFi channel or the encrypted IP packets captured on Ethernet or routers. Data collected could also be complete when capturing 100% of traffic going through or being sampled via various sampling methods.

*Data representation and preprocessing*: Various techniques have been employed to depict traffic. Moore et al. [70] present an extensive collection of characteristics derived from the packet size and inter-arrival time, TCP flags, port numbers, and IP addresses. Additional approaches take into account the domain names and the contents of protocol requests. Time series features are utilized for precise classification at a detailed level. An alternative method being suggested is the depiction of the patterns of communication between the entities involved using unconventional formats. Images have recently been used to depict traffic flows. The selection of data format is determined by the classification algorithm employed. For instance, the time series presentation is employed for classification based on Markov Models, while the image-based representation is utilized for classification based on deep learning, among other methods [71].

Before the network traffic data can be analyzed, it also needs to be preprocessed carefully. Data preprocessing is necessary because it will help clean up messy and/or inconsistent data, reduce data dimension and normalize data scale to improve analysis efficiency, reduce overfitting and bias in training, etc. [72] In summary, data preprocessing acts as an indispensable foundation for successful machine learning. It ensures the data are clean, prepares them for the model’s learning process, and ultimately improves the accuracy, efficiency, and generalizability of ML predictions and classifications.

*Feature extraction*: In machine learning, feature extraction is the process of transforming raw data into a set of more relevant and informative features. Feature extraction helps identify and extract the key features that actually matter for the prediction task by machine-learning models. It will greatly reduce data complexity and dimensions, selecting only those input dimensions that contain the most relevant information for solving the particular problem. Khalid et al. [73] provided a good survey on feature selection and extraction techniques in machine learning. The research conducted by Hakak et al. [74] is an example of work showing the importance of feature extraction techniques in identifying fake news more efficiently.

*ML model selection and refinement*: After data have been collected and processed, the next step is to use the right machine-learning model so that we can achieve the best traffic analysis and classification performance. Choosing the right machine-learning model is crucial for making the most of our data. Numerous ML models have been presented so far and utilized in various fields. Factors to consider include problem type, data characteristics, model complexity, interpretability, availability of resources, etc. [75].

ML model selection should be an iterative process that requires us to continuously refine our selected model and its parameter settings to better fit the target application and resources available. Therefore, as shown in Figure 2, this step has an iterative loop with the next step of ’ML model training’.

*ML model training*: Machine-learning training is the process of feeding data into a machine-learning model to help it learn and improve its ability to make predictions or classifications. The data are typically divided into three sets: the training set used to train the model, the validation set used to monitor performance during training and prevent overfitting, and the test set used for final evaluation after training is complete. Two main factors need to be considered in model training. First is the computational resource needed for training and the resource available for future real-world deployment. Second is the potential bias and fairness issues, especially when ML is used in social-impactful applications.

*Deployment*: Deployment is the process of making a trained machine-learning model accessible and operational in the real world, where it can be used to make predictions or classifications on new, unseen data. For network traffic analysis and classification, it can be deployed in a cloud platform on received real-time monitored data, on the premises if an organization deploys it on its own server due to various concerns, or on edge devices with limited connectivity or for strict real-time requirements [76].

*Classification*: For encrypted network traffic analysis, classification means to discover more insight information based on captured unknown encrypted traffic, such as the IoT device type or even working status, what applications generate the captured traffic, whether the encrypted data are audio, video streaming, or images, etc.

*Feedback and evaluation*: The classification results given by the machine-learning model can be further evaluated or verified based on either the classification outcome via other ways or human manual inspection. This feedback and evaluation will help users to understand better what are the potential weaknesses or problems with the applied machine-learning system and then redesign or refine all the previous procedure steps in the whole process shown in Figure 2 to improve classification performance.

## 5. Network Traffic Processing and Inspection

This section provides an overview of common methods and techniques for processing and inspecting network traffic that have been extensively utilized in the literature throughout the years. This section aims to facilitate the reader’s understanding of the process of network traffic processing and to highlight the issues that have arisen due to network encryption. Anomaly detection is a versatile technique that may be used in various fields, including network intrusion detection, to identify abnormal or suspicious activity and behavior.

A strategy based on anomaly detection estimates a model of “normal network activity”, and evaluates future activity on the network by comparing its probability to the learned model. By employing this method, it becomes feasible to differentiate between a usual and potentially harmful network activity as opposed to one that is harmless. This categorization is founded on specific heuristics or principles and seeks to identify instances of misuse or atypical conduct.

For an anomaly-based intrusion detection system to accurately detect suspicious traffic, it is essential for the system to be trained to understand typical system behavior. For instance, there are several ways available for detecting volumes. These techniques are used to monitor the traffic load of a network to detect abnormalities that cause major changes in traffic volume, such as flooding attacks.

Feature-based anomaly detection seeks to address the limitations of volume-based solutions by analyzing various characteristics of network data. The bulk of anomaly detection systems typically involves two phases, which are training and testing [77,78,79,80].

The first part involves constructing a profile of usual behavior, whereas the testing phase involves comparing current traffic with the model generated during the training phase. Machine or deep-learning techniques are mostly employed to detect anomalies. Nevertheless, anomaly detection frequently encounters challenges such as (i) elevated rates of false positives, (ii) the challenge of acquiring dependable training data, and (iii) the persistence of training data and the changing behavior of the system (Figure 3).

The multitude of protocols and the variety of data in modern connections make it more challenging to distinguish between normal and abnormal behavior. This complexity hinders the establishment of an accurate baseline and expands the potential for exploitation or zero-day attacks. While anomaly-based detection approaches can be highly efficient in the field of network security, signature-based techniques can analyze and examine a broader range of applications.

Signature-based network inspection is a method used to classify and characterize traffic, primarily using DPI techniques. DPI serves as a fundamental element in various network systems, including traffic monitors, classifiers, packet filters, network intrusion detection, and prevention systems.

Different layers of the OSI architecture utilize deep packet inspection in various network components [81,82,83,84]. In contrast to the initial stages of packet inspection, where DPI was limited to examining packet headers (such as proxies and firewalls), the present scenario necessitates the inspection of packet content across all layers of encapsulation due to the increased complexity and obfuscation of protocols.

Governments, Internet and Communications Service Providers (ISPs/CSPs), and other organizations largely depend on DPI technology to accurately monitor and analyze network traffic [85,86].

Utilizing DPI can enhance the quality of service (QoS) by discerning various types of content and streaming them in a tailored manner. Similarly, through the analysis of the contents of the incoming packets, one can detect and scrutinize suspicious and malicious activity.

In the field of network security, the use of signature-based network inspection techniques can be highly effective if security software developers are already aware of similar assaults. As previously said, DPI is the fundamental process used in typical applications within the field of network security, analytics, and other related areas. The need for advanced traffic analysis and the ongoing rise in network speeds have prompted extensive research to continuously develop innovative DPI methods.

Network intrusion detection systems have evolved into highly effective instruments for network administrators and security specialists in recent decades, aiding in the identification and prevention of a broad spectrum of threats. Snort and Suricata are widely used network intrusion detection system (NIDS) solutions that employ pattern matching and regular expressions to analyze network data. In contrast, Zeek/Bro utilizes scripts to facilitate automation, making it more convenient. The research community has also made endeavors to enhance the efficiency of NIDS by utilizing either standard hardware, such as graphics processing units (GPUs) and parallel nodes, or dedicated hardware, such as ternary content-addressable memories (TCAMs), application-specific integrated circuits (ASICs), and field-programmable gate arrays (FPGAs).

Nevertheless, these works can analyze network traffic that lacks encryption, as they extract significant data from the content of network packet payloads. Traffic classification and network analytics play a crucial role in ensuring quality of service (QoS) at both central network traffic intake points and end-host computers. To precisely identify the incoming traffic, a detailed analysis should be conducted, examining both the packet header and payload.

Most traffic categorization methods primarily focus on flow-based strategies, which try to categorize flows based on the application that generates them rather than individual packets. Several studies have suggested several techniques for determining the application linked to a traffic flow [87,88,89,90,91].

## 6. Strategies in Analyzing and Classifying Encrypted Network Traffic

Encrypting network traffic is crucial for ensuring data privacy and security, but it also poses challenges for analyzing and classifying that traffic for various purposes, such as network management, security monitoring, and traffic optimization. However, there are approaches to analyzing and classifying encrypted network traffic even without decrypting it fully, leveraging machine learning and deep-learning techniques.

In the following, we summarize strategies that can be utilized to perform effective analysis and classification of encrypted network traffic for various purposes while preserving data privacy and security.

Feature Extraction: Even though the payload of encrypted traffic is not accessible, features can still be extracted from the encrypted data. These features could include packet size, inter-arrival times, packet direction, protocol headers, and statistical properties of encrypted payloads.Dimensionality Reduction: High-dimensional feature spaces can be reduced using techniques like Principal Component Analysis (PCA) or autoencoders to capture the most relevant information while discarding noise.Model Training: Various machine-learning algorithms such as Random Forests, Support Vector Machines (SVM), or neural networks can be trained on the extracted features to classify different types of network traffic.Deep-Learning Approaches: Deep-learning models, especially Recurrent Neural Networks (RNNs) or Convolutional Neural Networks (CNNs), can be effective in learning patterns and representations from sequential or spatial data, which can be applied to encrypted network traffic analysis.Transfer Learning: Pre-trained deep-learning models on related tasks such as natural language processing or image classification can be fine-tuned for encrypted network traffic analysis, leveraging the learned representations.Anomaly Detection: Unsupervised learning techniques can be applied to detect anomalies in encrypted traffic by learning the normal behavior of the network and flagging deviations from it.Traffic Classification: Supervised learning techniques can be used to classify encrypted traffic into categories such as IoT devices, websites, or apps based on their behavioral patterns and communication protocols.Ensemble Methods: Combining multiple models or classifiers through ensemble methods like bagging or boosting can improve classification accuracy and robustness.Privacy Considerations: It is crucial to ensure that the analysis of encrypted traffic respects user privacy and regulatory requirements. Techniques like differential privacy or secure multiparty computation can be explored to achieve this.Continuous Learning: Given the evolving nature of network traffic patterns and security threats, continuous learning mechanisms should be employed to adapt the models to new data and emerging trends.

## 7. Limitations

*Networking dynamics:* network services such as asymmetric routing, NATing, and tunneling have an impact on the performance of traffic classifiers. Hence, it is imperative to account for these network dynamics while constructing a traffic classifier. The features must be selected in a manner that ensures they are not influenced by any of these network functions [92].

*Curse of dimensionality:* it refers to the challenge of efficiently classifying data, particularly in real-time processing, when speed is crucial. It is crucial to thoroughly analyze the time and processing complexity aspects of data preparation, data representation or feature computation, and classification computation overhead. Within this particular framework, three primary factors are crucial: memory capacity, computational intricacy, and processing duration. Developing a resilient yet precise classifier is crucial, yet efficiency is essential in certain scenarios, particularly for time-sensitive network services (such as intrusion detection) [93].

*Implementation challenges*: Despite the substantial research on ML traffic categorization, there are only a limited number of classification frameworks/tools available. Despite privacy issues, DPI continues to be widely employed. Furthermore, technical obstacles such as traffic velocity and large-scale data could potentially hinder the practicality of capturing and analyzing this traffic. Additionally, a constraint in real-world implementation is the need to continuously train the models, as well as adjust for new or unfamiliar traffic. Additionally, the optimization of model parameters, taking into account specific network features such as speed and fragmentation, may impose limitations on the performance of implemented models [94].

*Obfuscation*: The propriety of classifying and obfuscating information is a subject of controversy. From a privacy standpoint, classification is regarded as an intrusion that undermines user privacy. Nevertheless, in terms of network administration, attackers can employ obfuscation techniques to evade detection of their attacks. Classification is preferred by security and QoS applications in this situation, whereas obfuscation is favored by privacy [95].

*Traffic sampling*: The process of collecting a representative subset of network traffic data for analysis and monitoring purposes. An obstacle to the implementation of traffic classification applications is the need for high-speed capabilities in the core network. Due to the impracticality of extracting features from packets at a very high speed, traffic sampling is used as an alternative. Modifying traffic characteristics and statistical data can potentially detrimentally affect the accuracy of classification [96].

## 8. Recommendations and Future Perspectives

*Data collection:* Data are crucial for every machine-learning classification framework. It is crucial to train the model with representative data to uncover significant patterns that aid in accurately classifying unknown data. It is necessary to obtain data from various network conditions and locations, including edge and core points. Furthermore, it is important to carry out data labeling with precision.

Data diversity is crucial as it allows for the development of novel categorization applications using data from various apps and devices. Ultimately, the accessibility of public data is crucial for enhancing research in this field. Furthermore, the accumulation of traffic data over an extended period is crucial for analyzing traffic patterns. However, in this scenario, it is necessary to manage large volumes of data to handle the substantial quantity of traffic being generated [97].

*Model generalization:* refers to the ability of a machine-learning model to perform well on new, unseen data with minimal bias and volatility. To achieve generalization, it is necessary to evaluate the model using data obtained from various network contexts, a task that necessitates the participation of operators. DL-based classification algorithms are a suitable option in this context for developing classification models that may be applied broadly.

*Unidentified traffic identification:* traffic categorization aims to determine the precise type of traffic, including the application name or even the device type. Nevertheless, these classes are dynamic. Continuous advancements in the network domain result in the ongoing emergence of new applications, devices, and traffic types. Hence, the identification of novel traffic or malicious traffic is crucial to prevent mischaracterization.

In this scenario, it is necessary to assess the uncertainty of the classification model with respect to future forms of traffic, and additional methods need to be created for distinguishing between malicious and non-malicious anomalies. Unsupervised machine-learning approaches are required in this situation to identify unfamiliar traffic without any prior understanding of its properties or trends [98].

*Ensuring the classifier’s robustness:* Ensuring the classifier’s robustness against obfuscation or its ability to detect obfuscated traffic is crucial to prevent misclassification. Unsupervised learning is required in this scenario to identify unfamiliar traffic categories. Classifiers must undergo testing against various obfuscation strategies, and the selection of features should prioritize reduced susceptibility to obfuscation.

Accounting for adversarial learning attacks is crucial to identify novel or altered traffic in this particular scenario. Generative deep-learning architectures, now utilized for identifying fraudulent photos, news, and other forms of deception, can also be utilized for detecting unknown or aberrant traffic.

*Model update:* As previously stated, the network domain is seeing the emergence of new traffic types in a dynamic manner, particularly due to the connectivity of novel devices in the era of the IoT.

It is necessary to train the model again to accommodate new forms of traffic that have recently emerged. Creating machine-learning algorithms for real-time training is crucial for ensuring that traffic classifiers remain up to date. Reinforcement learning is essential for updating the trained model in QoS management based on user satisfaction feedback, as well as for managing security by considering the amount of positive and false alarms.

*Hierarchical classification:* It is crucial to achieve different classification objectives in network functions such as QoS management, firewall, intrusion detection, and application banning. Hierarchical classification allows for dynamic granularity levels, eliminating the need for multiple classification modules and training multiple classifiers [99].

When evaluating a series of machine-learning models to classify traffic based on many levels of classes, it might lead to a dynamic level of classification granularity. However, in this scenario, minimizing error propagation would significantly enhance the model performance, particularly when considering intrusion detection.

## 9. Conclusions

This paper presents a survey of classification approaches for encrypted network traffic for IoT devices, websites, and mobile applications. We reviewed various studies and methodologies in this field, highlighting key findings and trends. For IoT devices, classification methods focused on device fingerprinting and OS identification, aiding in device management and security. Website classification is aimed at detecting privacy leaks and malicious activities, such as financial fraud, through techniques like web fingerprinting and user action identification. Mobile application classification focused on network traffic analysis, QoE metric measurement, and anomaly detection, assisting in service optimization and security enhancement.

While these classification methods provide valuable insights, they also raise concerns about network security and user privacy. Countermeasures against these risks were also discussed. In conclusion, the classification of IoT devices, websites, and mobile applications is essential for network security and personalized services. However, careful consideration is required of its impact on security and privacy. Future research should focus on developing innovative approaches to address these challenges and ensure the safe and efficient operation of IoT, websites, and mobile applications.

## Figures and Tables

**Figure 1 sensors-24-03509-f001:**
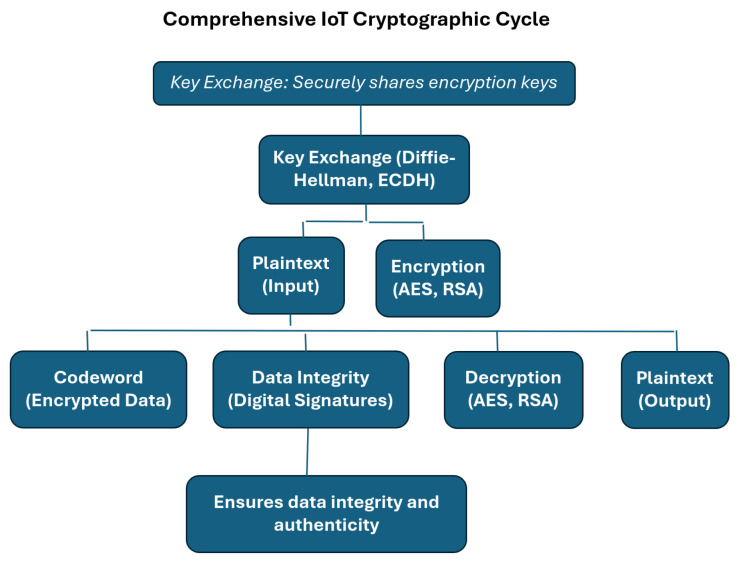
IoT crypto system shows how plain text is converted into codeword after using cryptographic algorithms.

**Figure 2 sensors-24-03509-f002:**
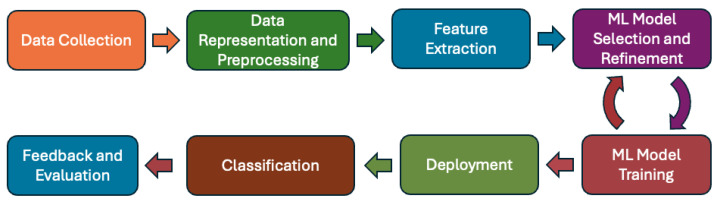
Procedure of typical encrypted network traffic analysis and classification based on machine learning.

**Figure 3 sensors-24-03509-f003:**
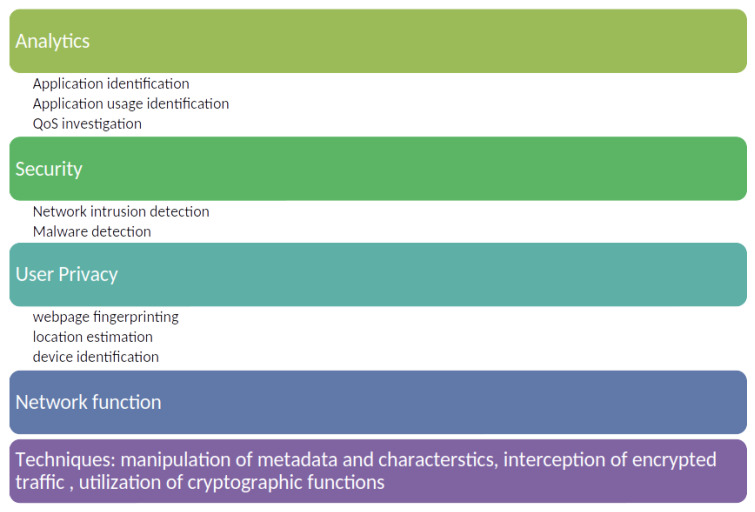
Encrypted traffic analysis and inspection phases.

**Table 1 sensors-24-03509-t001:** Comparison of Existing Surveys with Our Work.

Survey Paper	Year	Description
Unencrypted Traffic		
Buczak et al. [38]	2016	Introducing data mining and machine-learning techniques for cybersecurity intrusion detection
Jing et al. [39]	2018	Reviewing security data and analytical methods for detecting DDoS and Worm attacks
Fernandes et al. [40]	2019	Summarizing network data types and techniques for anomaly detection
Kwon et al. [41]	2017	Examining deep-learning methods applied to network anomaly detection
Encrypted Traffic		
Velan et al. [6]	2015	Summarizing approaches for analyzing encrypted traffic, mainly focusing on traditional machine-learning methods
Rezaei et al. [42]	2019	Reviewing deep-learning techniques for the classification of encrypted traffic
Conti et al. [7]	2018	Reviewing studies that focus on network traffic analysis targeting mobile devices
Shen et al. [43]	2023	Introducing machine-learning solutions for network traffic classification
Our Survey	2024	Providing a comprehensive survey on machine-learning-based methods for analyzing encrypted traffic, covering goals such as network asset identification, network characterization, privacy leakage detection, and anomaly detection.

## Data Availability

No new data were created or analyzed in this study. Data sharing is not applicable to this article.

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
