# Peer review of "Encrypted Network Traffic Analysis and Classification Utilizing Machine Learning"

_sensors, 2024, doi:10.3390/s24113509_

Round 1

Reviewer 1 Report

Comments and Suggestions for Authors

The document titled "Encrypted Network Traffic Analysis and Classification Utilizing Machine Learning" discusses the importance of understanding and classifying encrypted network traffic patterns, especially in the Internet of Things (IoT) context. The paper highlights the challenges posed by encryption on traditional traffic analysis methods and emphasizes the role of machine learning in extracting insights from encrypted traffic without accessing the content. Key points include the significance of network security, improved management of IoT devices, and the application of deep learning and machine learning in detecting anomalous or malicious behavior. The document also provides an overview of machine learning and deep learning techniques, their applications in various fields, and their role in enhancing the capabilities of IoT systems.

This paper presents complex concepts in an accessible and understandable manner, while also carving new paths of thought within existing academic discourse, undoubtedly making a significant contribution to discussions in the relevant field. However, despite its excellence in many aspects, there are still some areas that warrant further exploration and improvement.

1.      Is this article only proposing the concept of using deep learning and machine learning techniques to analyze and classify encrypted network traffic without completely decrypting data without showing experiments to verify this point?

2. Abbreviations (such as IoT) should be mentioned everywhere, or only once; please standardize the format.

3.      Line 221, there is no space between 67% and in.

4.      Line 252, the reference should be placed before the period.

5.      Is the reference in line 458 misplaced?

Author Response

1.    “Is this article only proposing the concept of using deep learning and machine learning techniques to analyze and classify encrypted network traffic without completely decrypting data without showing experiments to verify this point?”
Answer: We rewrite the abstract and introduction to clearly explain that this paper is a survey of research on analyzing and classifying encrypted network traffic by using various machine learning techniques. The encrypted network traffic cannot be decrypted by analyzers. Since this is a survey paper, we do not provide our own experimental evaluation but focusing on systematically summarizing and discussing prior research work on analyzing encrypted network traffic.

2.    “Abbreviations (such as IoT) should be mentioned everywhere, or only once; please standardize the format. Line 221, there is no space between 67% and in. Line 252, the reference should be placed before the period. Is the reference in line 458 misplaced?”
Answer: Thanks a lot for the detailed suggestions. In the revision we have revised the paper in most parts, and have fixed all the above-mentioned problems.

3.      "Line 221, there is no space between 67% and in."

Answer: We have revised most content in the original “Section 1.4. Mobile Applications” to become the new “Section 3.2. mobile devices and applications”. In the new version, the content mentioning the 67% has been removed. 

4.      "Line 252, the reference should be placed before the period."

Answer: Same reason as above. The reference in the original version’s line 252 has been removed because that part of content has been removed in the new revision.  

5.      "Is the reference in line 458 misplaced?"

Answer: In the new revision, this reference is [66] and is referred to in Line 175 in the correct format.  

Reviewer 2 Report

Comments and Suggestions for Authors

- Abstract is not well-structured. I suggest to the authors to clearly present the research goals, what they will research and how.
- Nowhere in Section 1 authors mention the structure of the paper, how introductory paragraphs explain the purpose of the research and similar.
- Several paragraphs in Introduction section (2nd, 3rd, 4th) explain general topic in IoT domain. I strongly suggest to authors to reformulate it and try to explain it in only 1 paragraph, but keeping smootly storytelling from the beginning of the 1 Section.
- Subsection 1.1 has no relation with the topic of the paper. Authors describe in general the machine learning, with terms that are already well-known. I was expected here to relate ML with the problem of security and which are the ways to utilize it for this purpose.
- Previously stated is also similar for subsection 1.2, 1.3, 1.4 and 1.5.
- In line 233, authors refers that they will talk in detail on some encryption methods in WiFi domain in section 1.3, while this line is in section 1.4.
- In subsection 1.6, authors nowhere discuss how the distribution of the packets affects the overal security in such system, which by my opinion must be considered. Some references which can help are:

- https://portal.sinteza.singidunum.ac.rs/paper/924
- Hu, Z., Odarchenko, R., Gnatyuk, S., Zaliskyi, M., Chaplits, A., Bondar, S., & Borovik, V. (2020). Statistical techniques for detecting cyberattacks on computer networks based on an analysis of abnormal traffic behavior. International Journal of Computer Network and Information Security, 12(6), 1.

- From Introduction section, sections 1.1 - 1.6 must be extracted and placed in separate section, for example Related Work. All of them must be completely reworked and text precisely related with the topic of the paper.
- Figure 4 in Section 2 is not described. For me is very unclear the presented steps in Section 2. More accurate will be if steps from Figure 4 is described here.
- Current Section 3 must be merged with new Section 2 (Related work) as proposed previously, while current Section 2 should became Section 3.
- Section 4 is redundant and parts of it should be integrated in new Section 3.
- Section 5 must be reworked. Very unclear it existence in the paper. What is the purpose of it? One idea here is to create it on the basis of the Figure 4., as a summary of it.
- Very bad formatting of Section 6. Authors nowhere discuss limitations inline with the aim of the paper, i.e. network traffic analysis and classification by using ML. How these limitations is related with your analysis of the research problem? How you come to them?
- Weak and poor Section 7. Recommendations and Future Perspectives must be given on the basis of the identified research gaps and problems and discussed limitations. Provided text is too general.
- Keeping on previously in mind, Section 8 must be thouroughly reworked and aligned with new text, results of conducted research and which novelty this research brings to the wider scientific community.

Author Response

1.    “Abstract is not well-structured. I suggest to the authors to clearly present the research goals, what they will research and how.” 
Answer: We revised the abstract completely and made sure that the abstract is clear and well-structured showing our research goals and contributions.

2.    “Nowhere in Section 1 authors mention the structure of the paper, how introductory paragraphs explain the purpose of the research and similar.”
Answer: We revised the introduction and explained the purpose of this research in both the abstract and the first several paragraphs in the Introduction section.  In addition, at the end of the first Introduction Section, we add a subsection to introduce the organization of this paper and what are the rest of the paper’s sections.  

3.    “Several paragraphs in Introduction section (2nd, 3rd, 4th) explain general topic in IoT domain. I strongly suggest to authors to reformulate it and try to explain it in only 1 paragraph, but keeping smoothly storytelling from the beginning of the 1 Section.”
Answer: Thanks for this suggestion. In the revision, we have used subsection 1.1 and its one paragraph to introduce classification of encrypted traffic in IoT application field. 

4.    “Subsection 1.1 has no relation with the topic of the paper
 Answer: In the revision, we have substituted the general introduction of machine learning in the original submission’s section 1.1 with a more focused introduction of utilizing machine learning in cybersecurity and in traffic classification.

5.    “Previously stated is also similar for subsection 1.2, 1.3, 1.4 and 1.5
Answer: We have revised this section to make sure that subsection 1.1 to 1.3 are used to introduce the three major applications (IoT, mobile, web) for classifying encrypted network traffic.

6.    “In line 233, authors refers that they will talk in detail on some encryption methods in WiFi domain in section 1.3, while this line is in section 1.4.”  
Answer:  Thanks for catching this writing error. We have revised and corrected this in the revision.

7.    “In subsection 1.6, authors nowhere discuss how the distribution of the packets affects the overall security in such system, which by my opinion must be considered. Some references which can help are….”
Answer:  Thanks for pointing out the two important related papers. In our revision, we have added and discussed the research in those two papers.

8.    “From Introduction section, sections 1.1 - 1.6 must be extracted and placed in separate section, for example Related Work. All of them must be completely reworked and text precisely related with the topic of the paper.”
Answer: In the revision we have moved related work discussion in the original Introduction section into the new “Section 2: related work”.

9.    “Figure 4 in Section 2 is not described. For me is very unclear the presented steps in Section 2. More accurate will be if steps from Figure 4 is described here… Current Section 3 must be merged with new Section 2 (Related work) as proposed previously, while current Section 2 should become Section 3..… Section 4 is redundant and parts of it should be integrated in new Section 3….. Section 5 must be reworked. Very unclear it existence in the paper. What is the purpose of it? One idea here is to create it on the basis of the Figure 4., as a summary of it
Answer: Thanks for pointing out the missing explanation of Figure 4 and excellent advice on how to rewrite the Section 5. In our revision, we have revised all the sections’ structure. Now the machine learning-based traffic classification procedure is depicted in Figure 2 (the old Figure 4 in first-round draft), and we use the new section, Section 4, to explain all steps in this procedure. 

10.    “Very bad formatting of Section 6. Authors nowhere discuss limitations inline with the aim of the paper, i.e. network traffic analysis and classification by using ML.”
Answer: We have revised this limitation section and focused on challenges and limitations of utilizing machine learning for network traffic analysis.

11.    “Weak and poor Section 7. Recommendations and Future Perspectives must be given on the basis of the identified research gaps and problems and discussed limitations. Provided text is too general.
Answer: In the revision we have updated and added some concrete discussion on identified research gaps and problems.

12.    “Keeping on previously in mind, Section 8 must be thoroughly reworked and aligned with new text, results of conducted research and which novelty this research brings to the wider scientific community
Answer:  We have revised the conclusion section completely. In addition, we have added Table 1 at the end of related work section listing the detailed contributions of our survey work in this research field compared with prior survey research.

Reviewer 3 Report

Comments and Suggestions for Authors

The current version has the following shortcomings.

1) What highlights have the current survey compared to other ones? I think the authors should add highlights to make readers understand clearly their contributions. Using a comparative table is preferred.

2) As a survey, the current work is superficial. Challenges and future directions that are the core of a survey should be extended to at least 4 pages, which will attract readers to make deeper research.

3) Some fresh and important references closely related to the current work should be discussed for completeness. For example, the work such as “Joint differential game and double deep Q-networks for suppressing malware spread in Industrial Internet of Things”, and so on.

Comments on the Quality of English Language

Minor editing of English language required.

Author Response

1.   “What highlights have the current survey compared to other ones? I think the authors should add highlights to make readers understand clearly their contributions. Using a comparative table is preferred.”
Answer:  Thanks for this excellent advice. In our revision, after we introduce the related work, we have added Table 1 at the end of related work section listing the detailed contributions of our survey work in this research field compared with prior survey research.

2.     “As a survey, the current work is superficial. Challenges and future directions that are the core of a survey should be extended to at least 4 pages, which will attract readers to make deeper research.”
Answer:  In our revision, we have expanded Section 7 (Limitations) and Section 8 (Recommendations and Future Perspectives) to discuss more in-depth on challenges and future directions in this area of research.

3.    “Some fresh and important references closely related to the current work should be discussed for completeness. For example, the work such as “Joint differential game and double deep Q-networks for suppressing malware spread in Industrial Internet of Things”, and so on.”
Answer:  Thanks for pointing out this great paper. We have added the paper in the reference ([16]) when introducing Internet-of-Things in Section 3.1.  

Round 2

Reviewer 2 Report

Comments and Suggestions for Authors

Manuscript is substantiallyyy improved according the reviewer comments. I suggest to accept it.

Reviewer 3 Report

Comments and Suggestions for Authors

The authors have made improvements to solve the concerns of reviewers. I think the current version is acceptable.